# Hydrodynamic Characterisation of the Inland Valley Soils of the Niger Delta Area for Sustainable Agricultural Water Management

**DOI:** 10.3390/s25144349

**Published:** 2025-07-11

**Authors:** Peter Uloho Osame, Taimoor Asim

**Affiliations:** School of Computing, Engineering & Technology (SoCET), Robert Gordon University, Aberdeen AB10 7GJ, UK; t.asim@rgu.ac.uk

**Keywords:** hydrodynamic characterisation, matric potential, soil subsurface flows, SWC, water content

## Abstract

Since farmers in the inland valley region of the Niger Delta mostly rely on experience rather than empirical evidence when it comes to irrigation, flood irrigation being the most popular technique, the region’s agricultural sector needs more efficient water management. In order to better understand the intricate hydrodynamics of water flow through the soil subsurface, this study aimed to develop a soil column laboratory experimental setup for soil water infiltration. The objective was to measure the soil water content and soil matric potential at 10 cm intervals to study the soil water characteristic curve as a relationship between the two hydraulic parameters, mimicking drip soil subsurface micro-irrigation. A specially designed cylindrical vertical soil column rig was built, and an EQ3 equitensiometer of Delta-T Devices was used in the laboratory as a precision sensor to measure the soil matric potential Ψ (kPa), and the volumetric soil water content θ (%) was measured using a WET150 sensor of Delta-T Devices. The relationship between the volumetric soil water content and the soil matric potential resulted in the generation of the soil water characteristic curve. Two separate monoliths of undisturbed soil samples from Ivrogbo and Oleh in the Nigerian inland valley of the Niger Delta, as well as a uniformly packed sample of soil from Aberdeen, UK, for comparison, were used in gravity-driven flow experiments. In each case, tests were performed once on the monoliths of undisturbed soil samples. In contrast, the packed sample was subjected to an experiment before being further agitated to simulate ploughing and then subjected to an infiltration experiment, resulting in a total of four samples. The Van Genuchten model of the soil water characteristic curve was used for the verification of the experimental results. Comparing the four samples’ volumetric soil water contents and soil matric potentials at various depths revealed a significant variation in their behaviour. However, compared to the predicted curve, the range of values was narrower. Compared to n = 2 in the Van Genuchten curve, the value of n at 200 mm depth was found to be 15, with *θ**r* of 0.046 and *θ**s* of 0.23 for the packed soil sample, resulting in a percentage difference of 86.7%. Additionally, n = 10 for the ploughed sample resulted in an 80% difference, yet *θ**r* = 0.03 and *θ**s* = 0.23. For the Ivrogbo sample and the Oleh sample, the range of the matric potential was relatively too small for the comparison. The pre-experiment moisture content of the soil samples was part of the cause of this, in addition to differences in the soil types. Furthermore, the data revealed a remarkable agreement between the measured behaviour and the projected technique of the soil water characteristic curve.

## 1. Introduction

Understanding how soils behave hydrodynamically is crucial for pushing the boundaries of agriculture. Understanding the different ways that water seeps into the soil is essential for irrigated agriculture. Because they regulate the infiltration process, soil water, and solute transport, soil hydraulic properties, such as the soil water retention curve (SWRC), diffusivity, and hydraulic conductivity functions, are essential in this regard [1,2,3]. Flood irrigation is the most often used technique in the Niger Delta region of Nigeria in sub-Saharan Africa, where farmers primarily rely on experience rather than scientific data. Currently, efforts are being made to switch to such irrigation techniques as drip irrigation that use less water. However, it is also critical to gain a better understanding of irrigation needs based on crop comfort zones and soil types. In order to give scientific support for the development of water-efficient irrigation techniques, this study aimed to shed light on the hydrodynamic behaviour of soils from two distinct locations in Nigeria: Ivrogbo and Oleh. In order to conserve water in agriculture, this study aimed to shed light on the intricate hydrodynamics of water flow through the soil subsurface [4]. Experimental measurements of soil hydraulic properties were essentially carried out using soil columns, either outdoors or in a laboratory equipped with instrumentation, to determine soil parameters such as temperature, water content, and soil pore pressure, as well as functions such as the soil hydraulic conductivity function, soil water characteristic curve, and soil water diffusivity. This is usually achieved by enclosing the soil column in a rigid, impermeable shell material for structural reasons and to prevent fluid loss. According to [5], since 1950, a great deal of work in the domains of hydrology, agriculture, and soil sciences has been published, most of which relies on the findings of soil column experiments.

Saturated hydraulic conductivity, Ks, can be determined in-situ using infiltration measurements in a variety of ways. Among the commonly utilised tools are the undisturbed soil core method (SCM), rainfall simulator (RS), tension permeameter (TP), constant-head well permeameter (CHP), single- and double-ring infiltrometers (SRI and DRI), and falling-head borehole permeameter. However, in-situ measurements of saturated hydraulic conductivity, Ks, using widely accepted instruments and techniques have frequently produced conflicting results. For example, Ref. [6] compared the estimates of the saturated hydraulic conductivity, Ks, of three traditional devices, namely the CSIRO version of a tension permeameter (CSIRO-TP), the Guelph version of a constant-head well permeameter (GUELPH-CHP), and a double-ring infiltrometer (DRI). The authors found that the estimates of Ks from the three devices were not very accurate when compared to benchmark values. They argued that the causes of the observed discrepancies still need to be looked into before using these techniques to confidently assess both field variability and local observations. In Ref. [7], the authors created a mobile, modified hood infiltrometer (MHI) design that allows the transient cumulative infiltration curve to be used to infer the hydraulic properties of soil. The commonly used devices offer significantly varying Ks estimations due to a number of factors, including flow geometry, sample size, soil type, soil conditions, and installation processes, according to an overall study of the aforementioned investigations. Soil column laboratory measurements of soil hydraulic properties include the steady-state method of determining the hydraulic conductivity of unsaturated soils involving the direct measurement of suction (negative pore water) values [8], measurement of essential hydraulic and electrical parameters of the regolith in crucial zones [9], study of the movement of water in an unsaturated soil column [4,10,11,12], automated soil water retention test device that controls water using a computer [13], experiments with irrigation on huge, undisturbed soil columns [14], preferential flow experiments on macroporous soils [15], and assessment of the soil pore water electrical conductivity (σp) change over time [16]. Even still, there has never been an attempt to standardize or gather the best practices for building soil columns, and a survey of the literature reveals a dizzying variety of technical methods. Understanding the soil water retention curve (SWRC), also called the soil water characteristic curve (SWCC), which shows the relationship between soil water matric potential (Ψ) and soil water volumetric content (θ), is crucial and necessary in order to describe and quantify the movement of water in the vadose zone, which is the area of the ground between the water table and the land surface, sometimes referred to as the unsaturated zone [17,18]. A typical profile of the soil water retention curve [19] is shown in Figure 1 below.

To determine the soil water characteristic curve, two major methods are commonly applied. The “direct method,” also referred to as experimental measurement, is the first. This can be accomplished through laboratory measurements or field (in-situ) measurements. The second strategy, referred to as the indirect method, estimates the soil water characteristic curve either directly or via the pedotransfer function process using data that are readily available to the public [9,20,21,22,23,24,25,26] or another procedure known as an artificial neural network [27,28]. In Ref. [29], the authors stated that the basic soil hydraulic property for predicting watershed runoff, scheduling irrigation, and determining the amount of water available for plants is the soil water retention curve. Soil hydraulic conductivity, moisture diffusivity or the coefficient of diffusion [8], and soil pore water electrical conductivity [16] are additional soil hydraulic properties.

Through a series of infiltration experiments employing gravity-driven water flow under ambient temperature conditions, this study measured the hydraulic properties of unsaturated soil in order to comprehend the intricate hydrodynamics of water flow through the soil subsurface. The soil water content, soil matric potential, and soil water characteristic curve are the hydraulic parameters of the soil that were measured and analysed in this study [4].

Soil water content, or the amount of water in the soil, can be quantified using either volumetric or gravimetric methods. The mass of water in the soil, defined as the difference between the damp and the dried soil at 105 degrees Celsius, is known as the gravimetric soil water content (θ_g_):(1)$$\theta_{g}=\frac{m_{W}}{m_{t}},$$
where $m_{t}$ is the total mass of the dry sample and m_w_ is the mass of the water in the sample. The amount of water per unit volume of soil is known as the volumetric soil water content.(2)$$\theta_{v}=\frac{V_{w}}{V_{s}},$$
where V_s_ is the total volume of the soil sample and V_w_ is the volume of water in the sample. While %vol is commonly used, m^3^.m^−3^ is the ideal unit for this ratio.

The quantity of water that is hydrogen-bonded to a plant’s matrix is known as its matric potential, and it is always negative to zero. The amount of water that is comparatively available and retained in the soil profile for plant uptake and utilisation is known as the soil matric potential (SMP). It illustrates the amount of energy plants need to separate water molecules from the particles. Since the water drawn to the soil matrix has a lower energy state than pure water, the matric potential is always negative. Using the terminology and notation of mechanics, the total potential *Ψt* of soil water is as follows [30,31]:(3)$$\Psi_{t}=-\int F_{g}\cdot dx-\int F_{op}\cdot dx-\int F_{p}\cdot dx-\int F_{ad}\cdot dx=-g\int dy-v\int d\pi -v\int dp-\frac{\varepsilon -1}{8\pi }v\int d{\left(\nabla \emptyset \right)}^{2},$$
where dx is the displacement vector; F_g_, F_op_, F_p_, and F_ad_ are the force vectors corresponding to four components of the total potential (gravitational, osmotic, pressure, and adsorptive potential, respectively); y is the direction pointing to the centre of the Earth; g is the acceleration due to gravity; v is the reciprocal density or the specific volume of water; π and *p* are the osmotic and gauge pressures, respectively; ε is the dielectric constant of water; and *ϕ* is the electric potential within the electrical double layer (EDL) formed at external surfaces of soil particles. The majority of the contemporary literature divides Ψt into three main components by combining the capillary pressure and adsorptive potential (the last two variables of Equation (3)) into a single term (the matric potential) [30,32,33](4)$$\Psi_{t}=\Psi_{g}+\Psi_{o}+\Psi_{m},$$
where Ψ_o_ is the osmotic potential brought on by the dissolved electrolytes in the water, Ψ_g_ is the gravitational potential resulting from the gravitational force field, and the matric potential that results from the interactions between soil and water is Ψ_m_. In [32,33], the authors contend that the adsorptive component is either purposefully disregarded due to oversimplification or mistakenly excluded in the conventional definition of matric potential as shown by Equation (4). A unitary definition of matric potential has been established in [31,34] utilising a mathematical description on the basis of gravimetric soil water content during either the wetting or drying phase as follows:(5)$$\Psi_{m}(\theta )=\Psi_{cap}(\theta )+\Psi (x,\theta )$$

Combining the last two terms of Equation (3) yields a similar result in Equation (5).

An essential idea in soil science and hydrology, the soil water characteristic curve (SWCC) offers vital information on the connection between soil water potential and soil water content. Understanding water movement in the vadose zone and how it affects different engineering and environmental processes depends heavily on this curve. Understanding plant–water connections, irrigation management, and soil moisture dynamics in agriculture all depend on the SWCC. In order to make well-informed decisions about crop selection, irrigation scheduling, and soil amendment techniques, it assists farmers in determining the available water-holding capacity of various soil types. Farmers can reduce crop water stress, maximise water use efficiency, and stop soil degradation from over-irrigation or water logging by using data from the SWCC. The soil water characteristic curve (SWCC) is uniquely defined by the relationship between the mass of moisture in a soil and the associated energy state, or suction, within the pore water [19,35,36,37,38]. The SWCC, which has a sigmoidal form for the majority of soils, as shown in Figure 1 above, describes the relationship between soil matric potential and moisture content [19,35,39]. The SWCC displayed in Figure 1 is described as follows. Its three distinct phases are the residual zone, transition zone, and border impact stage (saturated zone). The slope of the curve containing the inflexion point distinguishes two key components: the air entry value (AEV) suction and residual conditions, also referred to as residual suction or residual water content. The AEV, or bubbling pressure, is the suction value at which air begins to enter the largest voids in the soil. The residual suction value (RSV), sometimes referred to as residual soil suction, is the suction that matches the remaining moisture content at the residual condition. In order to experimentally investigate the profiles of soil water content and soil matric potential from unsaturated conditions, as well as to further investigate the soil water characteristic curve as a relationship between them and provide insight into their links to crops’ comfort zones in different soil types, this study was conducted to develop a soil column experimental setup for soil water infiltration experiments.

## 2. Materials and Methods

### 2.1. Soil Column Experiments

In order to accomplish its research objectives, this study employed the soil column approach. A specially designed soil column test rig was created and outfitted with the required tools to conduct soil hydrodynamic testing, as seen in Figure 2. As part of the investigation, a series of soil column water flow experiments were carried out using a rig that was designed and constructed. The soil column was positioned in the middle part of the experimental rig. The soil samples used in the experiment were included. Horizontal holes of 45 mm in diameter were drilled at 100 mm intervals down the 400 mm high soil column in order to install the sensors for measuring the water content and soil matric potential. The central section had a soil column that may have varying diameters. Initially, three WET150 sensors [from Delta-T Devices Ltd., Cambridge, UK] were integrated on one side of the 400 mm high soil sample column, spaced 100 mm vertically downward, and two EQ3 Equitensiometer sensors [from Delta-T Devices Ltd., Cambridge, UK] were inserted on the other side through 45 mm bored holes in the cylindrical pipe. Two EQ3 equitensiometers were placed 300 mm from the top, 100 mm apart, and the first one was placed 200 mm vertically downward. To find the amount of water in the soil before and after each infiltration experiment, the soil column and the sensors were weighed on a Seca weighing scale both prior to attachment to the GP2 data logger (from Delta-T Devices Ltd., Cambridge, UK) and again after logging was complete and the data logger was disconnected. Referring to Figure 2, the setup for the soil column infiltration studies conducted in the laboratory was the same for each sample. This process was used to calculate the flow Reynolds number and determine the hose’s flow velocity by considering the water released from the supply tank. For instance, using the ploughed soil sample, the experiment lasted 217 min, with data being recorded every minute. Even though 2140.71 g of water had been released from the supply tank, 312.99 g of water had been received in the receiver tank by the end of the experiment. Using the analytical formula (not presented here), the Reynolds number was calculated to be 4.194312518 based on the flow velocity of 8.38863 × 10^−5^ m/s, and the flow rate was calculated to be 1.64417 × 10^−7^ m^3^/s. Consequently, the flow was laminar and had a low Reynolds number, which made it perfect for drip micro-irrigation systems. Using the same procedure, the Renolds numbers for the packed soil, Ivrogbo soil, and Oleh soil were calculated to be 2.23678, 2.18625, and 2.55407, in that order.

Wall effect: in soil column water flow experiments, water migrates to the cylinder walls, resulting in preferential flow, which is a quicker flow around the walls brought on by a stronger attraction between the water and the cylinder wall. To avoid this, the cylinder’s inner wall was scraped with silicon gum. Water was also gradually fed into the soil column through a hose in the middle of the soil column to mimic drip micro-irrigation. This allowed the soil to become uniformly wet as it moved down the column. Furthermore, the geometry of the column was made to adhere to the 4-to-1 cylinder height to diameter ratio as per [40]. Additionally, the sensors were tightly fitted into the holes in the soil to minimize the preferential flow around them.

### 2.2. Data Logging and Monitoring Devices

The experiment’s data logging and monitoring were conducted using a GP2 Data Logger and Controller, a research-grade logger with intricate computed measurements and advanced feedback control. Pre-installed software on the PC was used to communicate with the data logger using the DeltaLINK 3.9 program software, which is a product of Delta-T Devices. A dataset import wizard was also installed on the computer in order to import DataLink datasets into a 32-bit Microsoft Excel spreadsheet. The dataset window was used to obtain and display all the data that had been saved in the GP2 data logger. When the dataset import wizard was used, these could be edited to open in Excel. They were saved to a PC dataset file. In order to record the soil water content, or θ, the data logger was connected to the sensors positioned in the soil column and connected to a PC running DeltaLINK version 3.9. This specific sensor, a multi-parameter Delta-T Devices Ltd., Cambridge, UK WET150 sensor, was suitable for usage in growing media such as soils and substrates. Over the whole range of 0 to 100%, the sensor converted the measured dielectric properties into water content with the greatest accuracy of ±3%. It also raised the bulk conductivity of the soil from 0 to 500 mS/m to 100%. The WET150 sensor measured the bulk electrical conductivity of soil (ECb) over a range of 0 to 2000 mS/m, with the best accuracy between 0 and 1200 mS/m ± (10 mS/m + 6%). It also calculated the electrical conductivity of the water contained in the soil’s pores, or pore water conductivity (ECp). As a result, the WET150 could be used with any third-party SDI-12-compliant logging or metering device, such as the Delta-T Devices UK GP2 data logger used in the experiment. An Equitensiometer (EQ3) sensor, which detects soil matric potential and temperature, was the main component of the soil matric potential recording system. The negative pressure, commonly known as suction, needed to extract water from the gaps between soil particles is known as the soil matric potential Ψ and has units of pressure, kPa. Numerous elements, including gravity, air pressure, osmosis, and the capillary action of soil particles, affect the water potential. The final component, called the soil matric potential Ψm, is highly dependent on the soil’s moisture content and ranges from 0 kPa at field capacity to about −1500 kPa at the permanent wilting point. It is an important indicator of plant water stress. The amount of water present and the makeup of the soil affect the observed value of the soil matric potential; however, temperature and salinity also play a role.

### 2.3. Testing Materials

In the inland valley of the Nigerian Niger Delta region, two monoliths of undisturbed soil samples were collected from fallow areas using the method of [14]. One was from Ivrogbo, close to a branch of the Niger River (5.41872 N, 6.34359 E). which had been used for agriculture in one form or another for over 15 years. However, when the sample for the experiment was taken, it was covered in weeds. The other was taken from a farm garden in Oleh (5.46186 N, 6.20624 E) that was left fallow and weed-grown at the time the experiment sample was taken. This area had also been used for agricultural purposes in one way or another for more than 20 years. The two areas have an equatorial climate with two distinct seasons and heavy humidity and rainfall for the majority of the year. To make the land cultivable all year round, however, an irrigation program outlined in this paper can be used throughout the dry season. Weeds were removed from the top in both cases, and core samples were collected using cylindrical PVC pipes with internal diameters of 101.6 mm and heights of 400 mm. The samples were then taken to the laboratory for initial sieve analysis in order to ascertain the bulk density, Atterberg limits, and soil particle size. Figure 3 and Figure 4 display the specifics of the initial investigation. At Robert Gordon University in the Garthdee neighbourhood of Aberdeen, United Kingdom (56.123136 N, 2.134911 W), a third soil column experiment sample was collected from a garden close to the River Dee. The region has an oceanic climate with extremely cold temperatures throughout the year. However, it has four clearly defined seasons of summer in June, autumn in September, winter in December, and spring in March. Based on the grain size distribution, it was determined to be silty sand. The sample was damp-packed [41,42] in a transparent acrylic PVC cylindrical pipe that was 400 mm high and had an internal diameter of 139.7 mm. The pipes in all the cases had 45 mm diameter holes that were drilled vertically downward at 100 mm intervals for the placement of sensors in the laboratory, as shown in Figure 5. This process involves mechanically packing small volumes of damp soils into the soil column [43]. After drying for about six months in the laboratory under standard pressure and temperature settings, the same packed soil sample from Aberdeen was disrupted to simulate ploughing in the fourth experiment because repeating the experiment immediately after the first run generated a matric potential very close to saturation, with a very high water content. Furthermore, preferential flow would be established if dried in the natural environment, because the structure of the matrix would also change.

### 2.4. Experimental Procedure and Sequence

Before logging, the weight of the soil column was recorded by weighing the dry, unsaturated soil with sensors on a Seca weighing scale. The weight of the soil column with the sensors was also monitored following the experiment (logging) in order to track the amount of water retained in the soil over the course of the experiment. Furthermore, water was poured into the rig’s upper supply tank, and the electronic laboratory scale underneath it was used to determine the tank’s weight. The weight was also recorded after each experiment to determine how much water entered the soil column. 

Using an electronic weighing scale beneath it in the laboratory, the lower part of the rig’s receiver tank was also weighed before and after each experiment in order to determine the water flux. By selecting an EQ2x sensor type measurement and inserting the unique lookup table that came with each Equitensiometer sensor, the EQ3 GP2 logger channel was formed in DeltaLINK. The GP2 multifunction program in the DeltaLINK 3.9 software was used to create the experiment program, which was then verified, saved, and sent to the data logger. The experiment began as soon as the logging was started. 

The faucet in the supply tank in the upper part of the rig was opened to provide the impression of drip micro-irrigation. This allowed water to drip into the soil in slow droplets through the hose. The flow rate was determined using the amount of water released as recorded over time by the electronic laboratory weighing scale. When water began to build up in the receiver tank, the dataset was analysed using the dataset window in the DeltaLINK 3.9’s GP2 logger interface, as shown in Figure 6. The pressure value was shown to rise from a negative value to zero and then move in the direction of a positive value. Logging was then stopped, and the soil column was removed from the logger and weighed again after the dataset was imported into an Excel file using the dataset import wizard.

### 2.5. Van Genuchten Correlation of the Soil Water Characteristics Curve

Van Genuchten (1980) suggested a method based on fitting soil water characteristic data using van Genuchten’s (1980) equation [19]. The van Genuchten (1980) equation [19] for the soil water characteristic of unsaturated soils is presented in the following equation:(6)$$\theta =\theta_{r}+\frac{\left(\theta_{s}-\theta_{r}\right)}{{\left[1+\left(\alpha \psi^{n}\right)\right]}^{m}}$$
where θ is the volumetric water content, $\theta_{s}$ is the saturated volumetric water content, $\theta_{r}$ is the residual volumetric water content, α and n are the curve fitting parameters, and m = 1 − 1/n.

Further, a is the shape parameter related to the air entry value: a = 1/hd × (2^(n/n−1)^)^(1/n)^, where hd is the minimum drainage pressure value. The pressure head at which air begins to enter the soil pores during drainage is the pressure at which the soil starts to desaturate from a fully saturated state.

## 3. Results and Discussion

### 3.1. Results of Sieve Analysis

Figure 3 displays the findings from the sieve analysis of the Ivrogbo and Oleh samples. The particle diameter of the Oleh specimen was 0.6 mm, whereas the diameter of the Ivrogbo sample was 0.3 mm, as the diameter is defined as the diameter for which the percentage passing in the sieve analysis is 90%. Figure 4 displays the test output graph for the Atterberg limit test results for the soil samples from Ivrogbo and Oleh. For both the Ivrogbo and Oleh soil samples, the bulk density test produced an average bulk density of 1.64 g/cm^3^, as shown in Table 1. Aggregation includes enhanced resistance to compaction and particle adhesion. The bulk density of sandy soil is normally 1.50–1.70 g/cm^3^. Consequently, porosity ranges from 0.43 to 0.36. The bulk density of clay soil is normally between 1.1 and 1.3 g/cm^3^. As a result, the porosity ranges between 0.58 and 0.51. Based on the bulk density of 1.64 g/cm^3^, the porosity of the soil samples from Ivrogbo and Oleh was determined to be 0.36. According to the American Association of State Highway and Transportation Officials (AASHTO) and the Unified Soil Classification System (USCS) [44,45], the Oleh test material was a sand–clay mixture, whereas the Ivrogbo test sample was identified as silty clay with low-to-medium plasticity based on the sieve analysis and the gaseous analysis (Table 2).

### 3.2. Measurement of the Soil Water Content

The soil water content was measured using the WET150 at 100 mm depth, 200 mm depth, and 300 mm depth vertically down the soil column for the four soil samples (Ivrogbo, Oleh, packed and ploughed Aberdeen soil specimen). The WET150 sensor automatically recorded the initial water content of each sample at the position where the sensor was placed at the beginning of the logging. Figure 7, Figure 8 and Figure 9 show the domain-specific soil water content at 100 mm depth for the ploughed, packed, Oleh, and Ivrogbo soil samples. With 17.5% water content at 100 mm depth, the Oleh soil sample was found to be the wettest prior to the experiment, whereas the Ivrogbo sample had 11.3%. The water content of the Aberdeen packed and ploughed samples was ca. 2.5 percent. However, as the infiltration tests progressed, it was found that every soil sample reached a constant water level, with the Aberdeen packed sample having the lowest water content and the Ivrogbo sample having the highest. Soil properties, including texture and structure, control size distribution, which in turn controls total water storage, available water holding capacity, and water circulation in the soil. Although changing the texture of the soil usually does not result in better plant–water connections, adding organic matter to the soil can enhance meso- and macro-porosity, which helps with free drainage and increases the amount of water that plants can hold. Understanding how soil and water interact is crucial for the majority of land use decisions [46]. For example, the Oleh sample had 33.2% water content at 100 min into the trial, whereas the Ivrogbo soil sample had 35.7% water content. The water content of the ploughed sample was 32.7%, whereas the water content of the packed sample was 23.6%. This was because the packed sample’s silty sand texture made it coarse-grained, but the Ivrogbo soil’s silty clay with low-to-medium plasticity texture rendered it fine-grained. Although sandy soils have lower accessible water values, fine-grained soils have more available water due to their larger pore count [47]. Because the ploughed sample had become finer-grained after the stones were removed from the packed sample, it was found to have a higher water content than the packed sample. After 110 min, it was observed that the Oleh, Ivrogbo, ploughed, and packed samples had the highest and lowest water contents, respectively, at a depth of 200 mm. At a depth of 300 mm, the same order of magnitude was visible. It demonstrated that, at that section, the Oleh sample had finer grains than the Ivrogbo sample vertically downward. It was found that repeating the infiltration experiments and calculating the average values for the matric potential and soil water content for the several runs on each sample was not appropriate in this investigation. It is evident that the clay concentration of the Ivrogbo and Oleh soils causes them to solidify when dried after the initial run.

### 3.3. Measurement of the Soil Matric Potential

The domain-specific profiles for the matric potential for the packed soil sample, the ploughed soil sample, the Ivrogbo soil sample, and the Oleh soil sample at 200 mm depth and at 300 mm depth are shown in Figure 10 and Figure 11. Because the ploughed soil sample was initially drier than the packed soil sample, it had a higher negative matric potential at 200 mm depth, but within the first 90 min, it decreased to about −8 kPa from −710 kPa, giving a slope of roughly 7.8 kPa/min, and the packed sample increased from −436 kPa to −490 kPa, giving a negative slope of −0.6 kPa/min. Both samples had low Reynolds numbers under laminar flow conditions. However, when the stones were removed from the packed sample to create the ploughed sample, the soil texture changed. The ploughed sample became siltier, more uniform, and had a larger field capacity, whereas the packed sample became sandier. Under the same laminar flow circumstances, this caused the ploughed sample’s matric potential to fall more quickly than that of the packed sample. It was found that the packed sample took almost 200 min to reach a stable matric potential, but the ploughed sample reached a constant value in roughly 90 min. Although their laminar flow conditions were identical, the pre-experiment soil in Oleh was wetter. However, in 90 min, with a gradient of 0.95, the matric potential of the Ivrogbo soil dropped from −98.2 kPa to −4.6 kPa before stabilising, while in 90 min with a gradient of 0.2, the matric potential of the Oleh soil dropped from −20.1 kPa to −2.2 kPa before stabilising. The ploughed soil sample was instantly at the permanent wilting threshold with a matric potential of −1850 kPa, whereas the packed soil sample had an initial matric potential of −28.1 kPa at 300 mm deep (Figure 11). Furthermore, the Ivrogbo soil sample began at −539.2 kPa, whereas the Oleh soil sample began at −19.3 kPa. After 125 min, their matric potentials were compared, and it was discovered that the ploughed soil sample’s matric potential had decreased with a gradient of about 14, while the packed soil sample’s gradient was negative at −0.0088. With a gradient of 4, the matric potential of the Ivrogbo soil sample dropped from −539.2 kPa to −17.4 kPa, whereas the matric potential of the Oleh soil sample dropped from −19.3 kPa to −17.4 kPa with a gradient of 0.02. As a result, there was agreement between the matric potential profiles of the 200 mm depth and the 300 mm depth. This demonstrated that the kind or texture of the soil had an impact on the profile of the soil matric potential when water was added [48,49,50,51]. This study found that the packed soil, which was assumed to be silty sand, was less silty than the ploughed soil. The Oleh soil sample’s sand–clay blend had the greatest field capacity. Next came the Ivrogbo soil sample, which is a blend of silty clay with low-to-medium plasticity, the ploughed soil sample, and finally, the packed soil sample. As also mentioned in [52], it showed that the soil samples with fine-grained textures had a greater field capacity than the soil samples with coarse-grained textures.

### 3.4. Measurement of the Soil Water Characteristic Curve

During the infiltration experiments of the different soil samples—that is, the packed soil sample, the ploughed soil sample, the Ivrogbo sample, and the Oleh sample—the relationship between the soil water contents and the soil matric potentials was plotted for 200 mm and 300 mm depths and compared, as shown in Figure 12. Furthermore, a comparison was made between the model of the van Genuchten curve and the soil water characteristic curve at a depth of 200 mm, as presented in Figure 13 and Figure 14. The soil water characteristic curves of unsaturated soils are greatly influenced by a number of variables, such as particle size, pore structure, initial water content, soil dry density, and soil texture [52]. Higher available water values are found in fine-textured soils, whereas lower available water values are found in sandy soils [47,52]. Even though the four soil types were exposed to similar laminar flow conditions, Figure 12a shows that their soil water characteristic curves differed significantly at a depth of 200 mm. While the Ivrogbo soil sample’s soil water content increased from 8.8% at a matric potential of −98 kPa to 35.3% at a matric potential of −4.4 kPa over a range of approximately 100 min, the packed soil sample’s soil water content increased from 4.7% at a matric potential of −436.5 kPa to 11.2%, but the matric potential increased to −485.2 kPa. This was due to the fact that the Ivrogbo soil had a finer grain size and a greater capacity to hold water than the packed soil sample, which had coarser grains. In contrast, the ploughed soil sample’s soil water content rose to 30.4% at a matric potential of −7.9 kPa from 3.1% at a matric potential of −710 kPa. This was due to the fact that the removal of stones and ploughing of the soil changed the texture of the soil and made the grains finer, which increased the soil’s capacity to store water. Because of the fine-grained clay composition and the fact that the sample was already extremely wet prior to the experiment, the Oleh soil sample’s soil water content rose from 15.9% at a matric potential of −20.1 kPa to 40.3% at a matric potential of −2.6 kPa. In Figure 12b, the same pattern was observed at a depth of 300 mm. While the packed sample’s water content increased from 6.1% at a matric potential of −28.1 kPa to a water content of 6.2% with the matric potential having increased to −29.2 kPa, the Ivrogbo soil sample’s water content increased from 10.8% at a matric potential of −539.2 kPa to 30% at a matric potential of −17.5 kPa in 130 min. This was due to the fact that the packed sample was fine-grained, which improved its water-holding capacity, while the latter was coarse-grained. With a matric potential of −1850 kPa and a water content of 4%, the ploughed soil sample was initially at the permanent wilting point. However, over the same time period, the water content rose to 23.8% at a matric potential of −58.8 kPa because the texture changed due to the removal of stones and the ploughing conditions, and its water-holding capacity increased as well. Due to the fine-grained sand–clay mixture and pre-experiment moisture, the water content of the Oleh soil sample rose from 28.2% at a matric potential of −19.3 kPa to 35.5% at a matric potential of −16.8 kPa at a depth of 300 mm.

At 300 mm depth, the water content for the Oleh sample was 28% when the matric potential was −20 kPa, and the water content of the Ivrogbo sample was 28% when the matric potential was also −20 kPa. However, the water content of the packed sample was 31% when the matric potential was −20 kPa at that depth, while the water content for the ploughed sample was around 30% when the matric potential was −20 kPa. With these known conditions of the soil characteristics of the different samples, a smart drip micro-irrigation system can be installed with crops of known comfort zones with a minimum threshold of −20 kPa matric potential. We took the case of the orange tree where the crop is most comfortable within a range of matric potential from −20 kPa to −100 kPa. With a smart drip micro-irrigation system installed in an orange orchard planted in the Oleh soil sample, drought will trigger smart irrigation when the water content falls below 16%. With smart drip irrigation in the Ivrogbo soil for the orange orchard, it will be triggered when the water content falls below 20%. The system will be triggered when the water content falls below 22% in the packed sample and below 24% in the ploughed sample. This means that the Oleh sample presents the most water savings in this scenario.

Figure 13 and Figure 14 display the measured and anticipated soil water characteristic curves for the ploughed and packed soil samples. The measured SWCC for the infiltration experiment at 200 mm is displayed in (b), whereas the predicted van Genuchten’s SWCC is displayed in (a) for each case. The soil matric suction is noticeably very high at the lowest value of the soil water content for the expected values, suggesting that the soil was quite dry and that the water content was residual. In the residual zone, the matric pressure head decreased at very low constant soil water content; in the transition zone, the matric pressure head reduced at a significantly slower pace when the water content rose rapidly. At the upper end of the 50% range, the water content stabilised and formed a vertically upward and downward S-shaped (sigmoid) curve. At this time, the curve was getting close to saturation. A low matric potential value of −0.1 kPa at a high volumetric water content of 29% and a high matric potential value of −436.5 kPa at a low volumetric water content of 4.7% are also displayed in the packed soil sample’s soil water characteristic curve. However, compared to the expected curve, the range of results was lower. The value of n was noticed to be 10, with $\theta_{r}$ = 0.046 and $\theta_{s}$ = 0.23. When compared to n = 2 in the van Genuchten curve, this resulted in an 80% difference in percentage. In addition to variations in soil type, this was partly caused by the soil sample’s pre-experiment moisture level. With a feature of an S-shaped curve (sigmoid curve), the soil column experiment for the packed soil sample also accurately defined the characteristics of the residual zone, transition zone, and saturation zone. These results were in reasonable agreement with the literature, as demonstrated in [35]. Similarly, the value of n was noticed to be 15, with $\theta_{r}$ = 0.03 and $\theta_{s}$ = 0.23 in the ploughed sample (Figure 13). This gave a percentage difference of 86.7% when compared to n = 2 in the van Genuchten curve. For the Ivrogbo sample and the Oleh sample, the range of the matric potential was relatively too small for a comparison. The pre-experiment moisture content of the soil samples was part of the cause of this, in addition to differences in the soil types.

## 4. Conclusions

A thorough and in-depth analysis was carried out since the goal of this study was to improve understanding of the complex hydrodynamics of water flow processes through the soil subsurface in order to conserve water in micro-irrigation activities. A real-world set of experiments was conducted in the laboratory to perform gravity flow on packed and ploughed soil samples from Aberdeen and monoliths of undisturbed soil samples from Ivrogbo and Oleh, two distinct regions from the inland valley of the Niger Delta region of Nigeria, using a conceptualised, designed, and built soil column test rig.

Measurements of the soil water content, soil matric potential, and soil water characteristic curve were among the methods used to examine the behaviour of soil hydraulic properties. The following deductions were made:

When compared to the profiles for the packed soil column, the undisturbed monolith of the soil column typically indicated non-equilibrium flow or preferential flow behaviour, while the ploughed soil showed uniform flow. At a low soil water content, high matric suction values were noted, whereas at a high soil water content, low matric suction values were observed. The saturation zone was linked to a high water content, whereas the residual zone was linked to a low water content. Higher accessible water values were found in samples with finer textures, whereas lower available water values were found in samples with coarser textures. The water-holding capacity adjusted to maintain a higher value of accessible water for the finer-grained texture as the soil texture changed vertically downward. However, a plant comfort zone could not be accurately determined by the soil water content. It is necessary to determine the soil texture and the matric potential to determine the plant comfort zone and match it with the water content. This could serve as a guideline for systems that use micro-irrigation. The matric potential and field holding capacity are significantly impacted by the texture of the soil. The van Genuchten model of the soil water characteristic curve was used to validate the experimental data. There was a notable difference in the behaviour of the four samples when comparing their volumetric soil water contents and soil matric potentials at different depths.

## Figures and Tables

**Figure 1 sensors-25-04349-f001:**
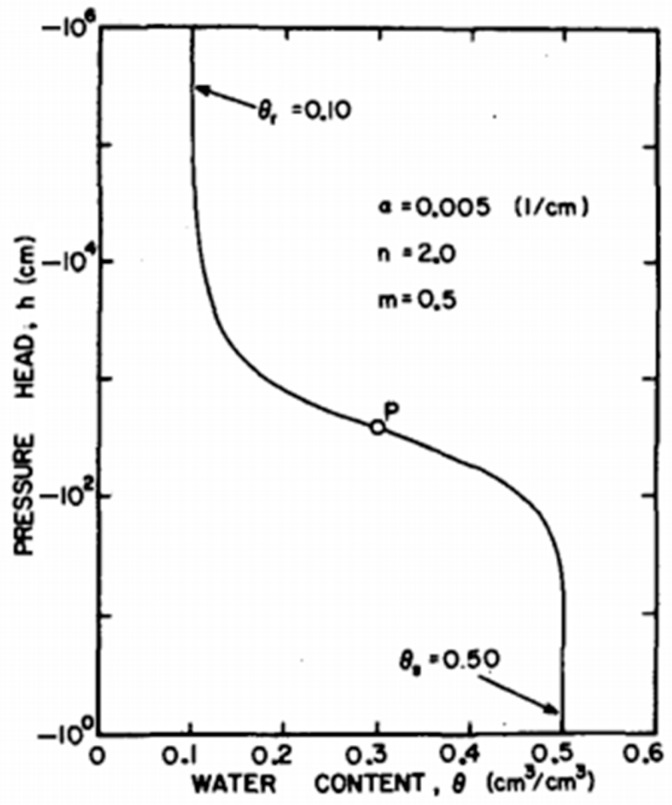
Typical profile of the soil water retention curve [19].

**Figure 2 sensors-25-04349-f002:**
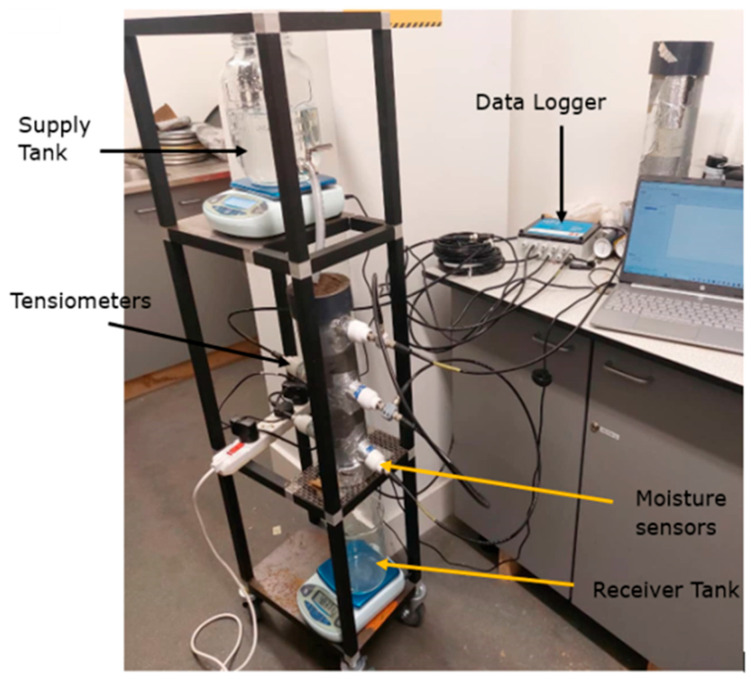
Constructed experimental setup that served as the workstation for the soil column experiment [4].

**Figure 3 sensors-25-04349-f003:**
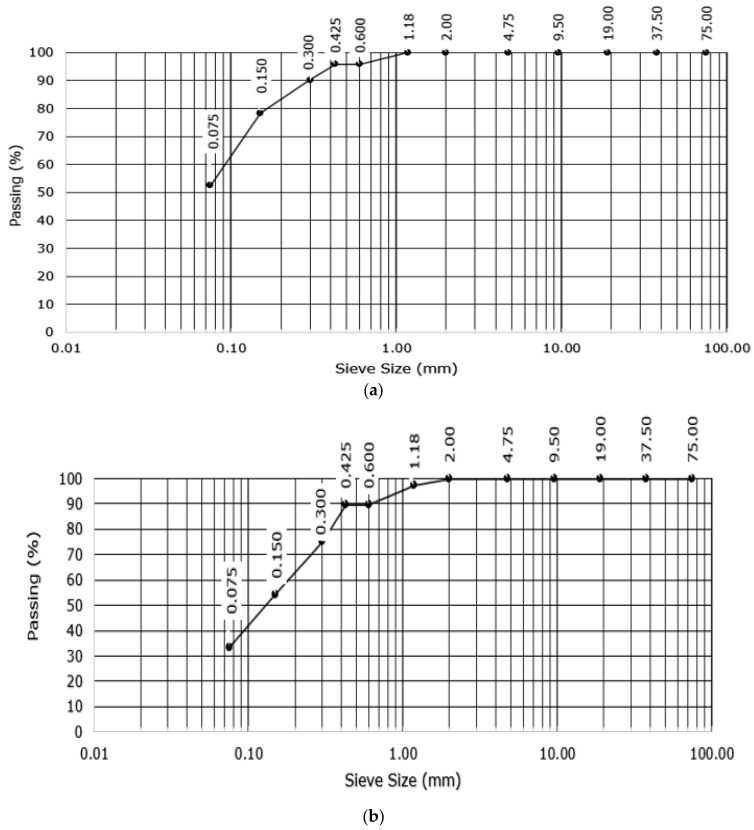
Particle size distribution: (**a**) Ivrogbo sample; (**b**) Oleh sample.

**Figure 4 sensors-25-04349-f004:**
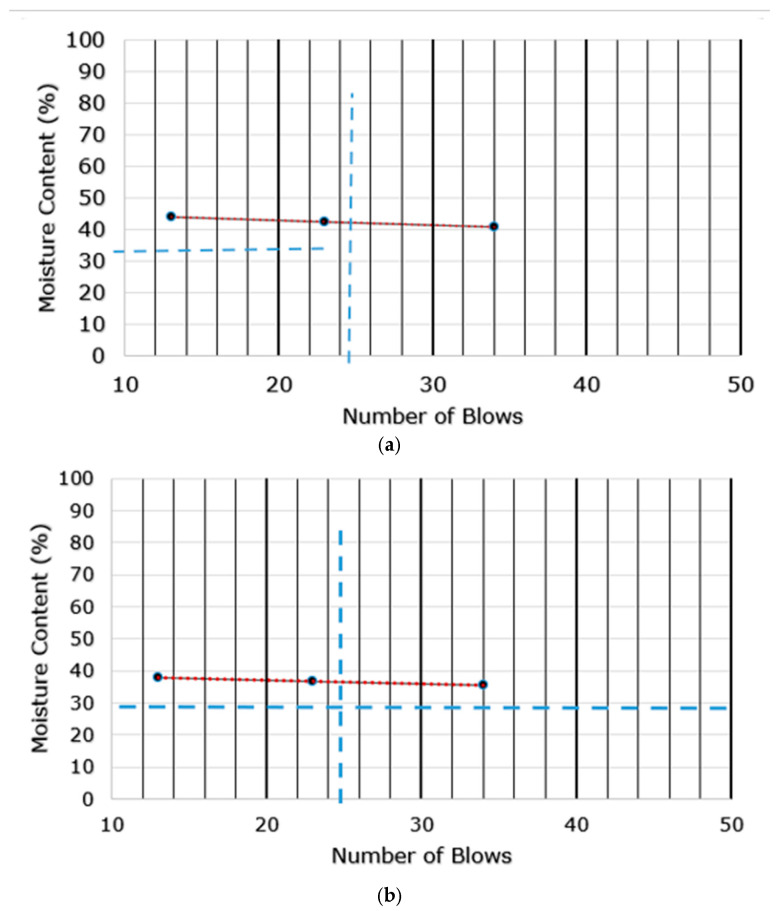
Atterberg limit test output graphs: (**a**) Ivrogbo sample; (**b**) Oleh sample.

**Figure 5 sensors-25-04349-f005:**
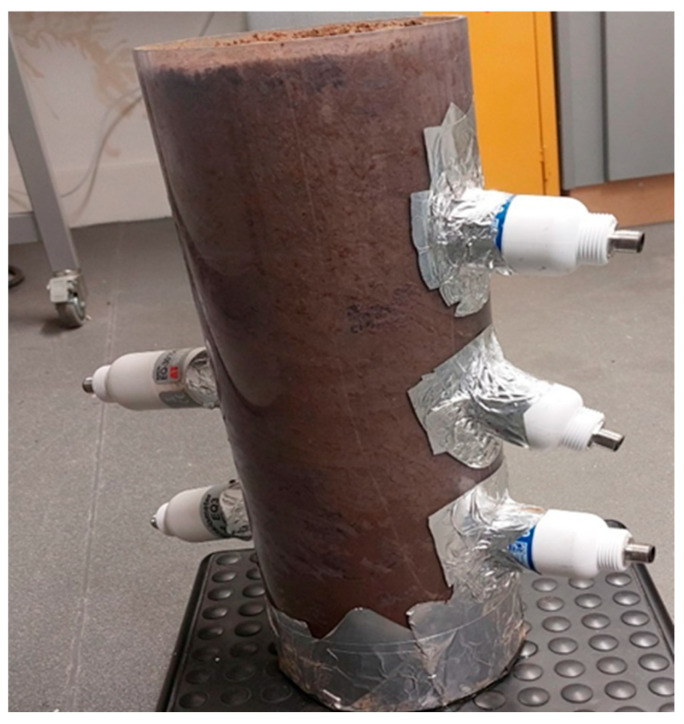
Packed soil sample.

**Figure 6 sensors-25-04349-f006:**
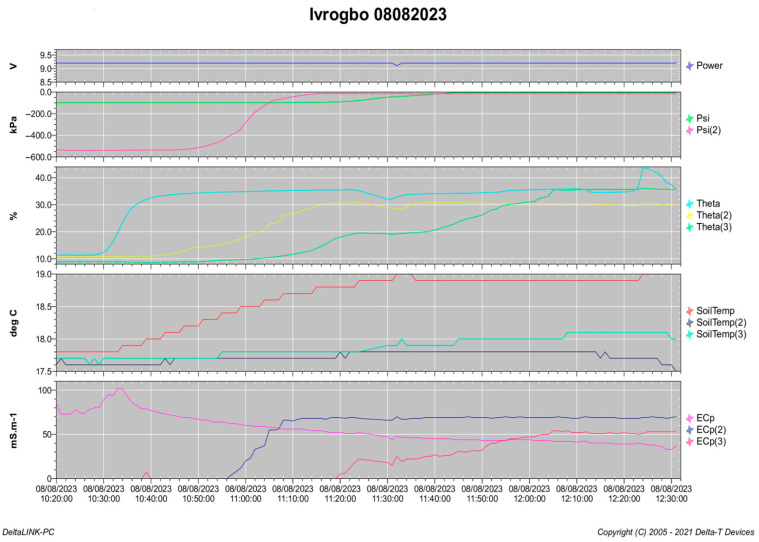
DeltaLINK dataset in the GP2 data logger interface on the computer.

**Figure 7 sensors-25-04349-f007:**
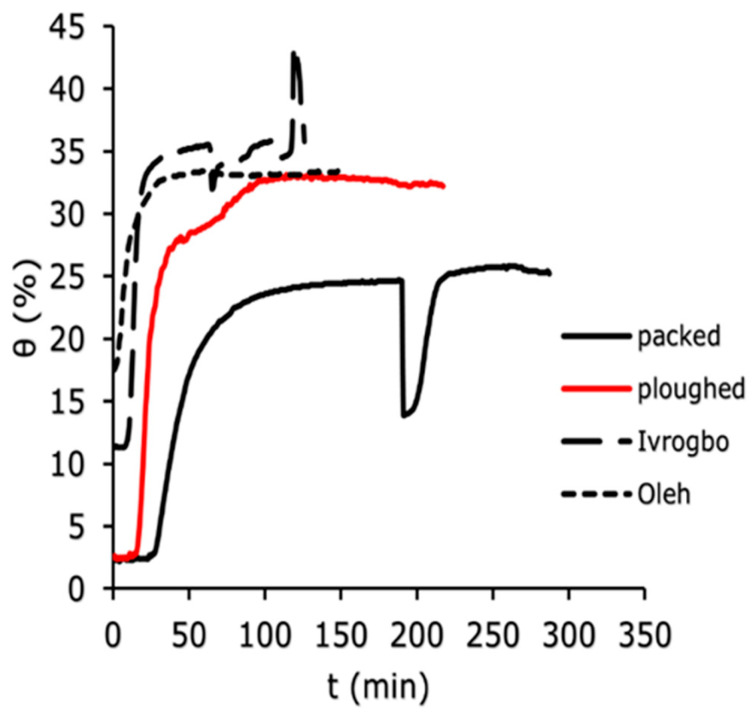
Domain-specific soil water content for the different soil types at 100 mm.

**Figure 8 sensors-25-04349-f008:**
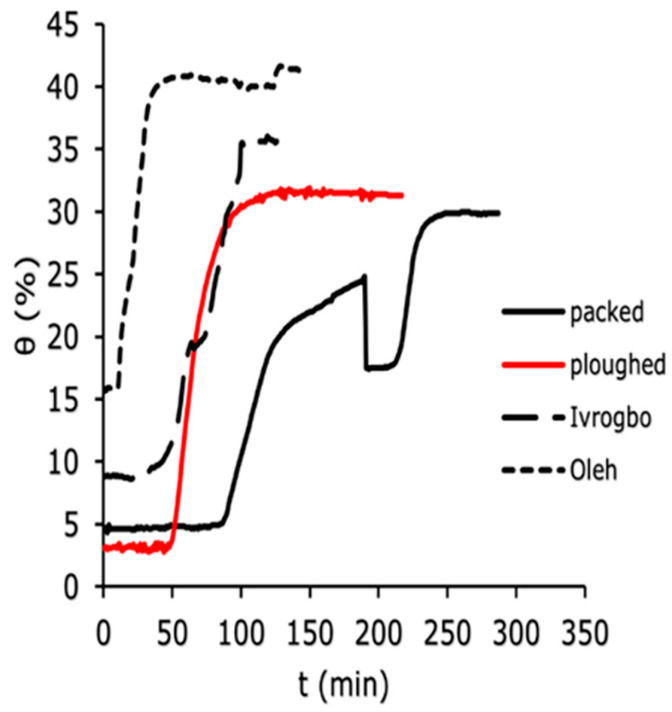
Domain-specific soil water content for the different soil types at 200 mm.

**Figure 9 sensors-25-04349-f009:**
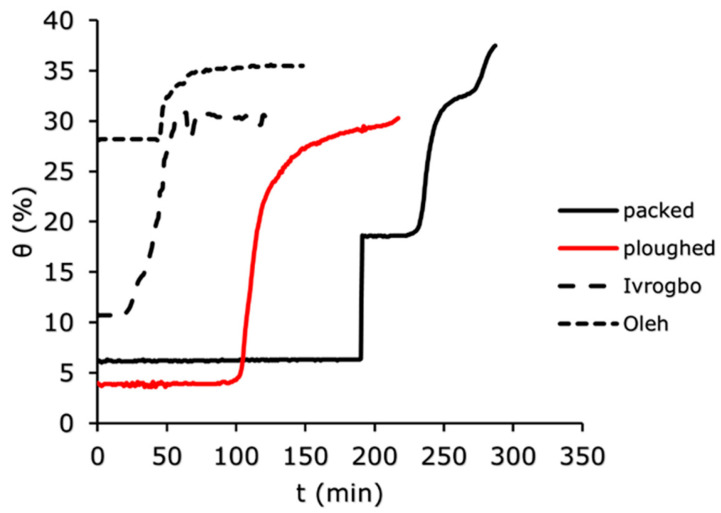
Domain-specific soil water content for the different soil types at 300 mm.

**Figure 10 sensors-25-04349-f010:**
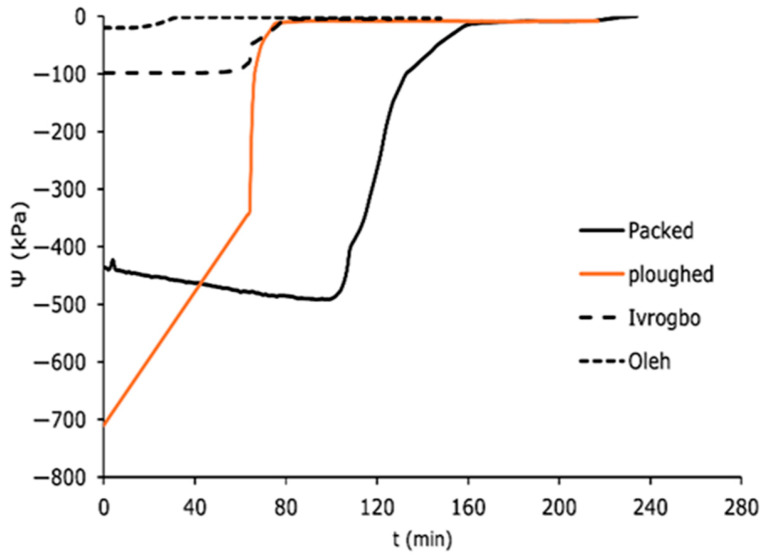
Domain-specific soil matric potential in the soil types at 200 mm.

**Figure 11 sensors-25-04349-f011:**
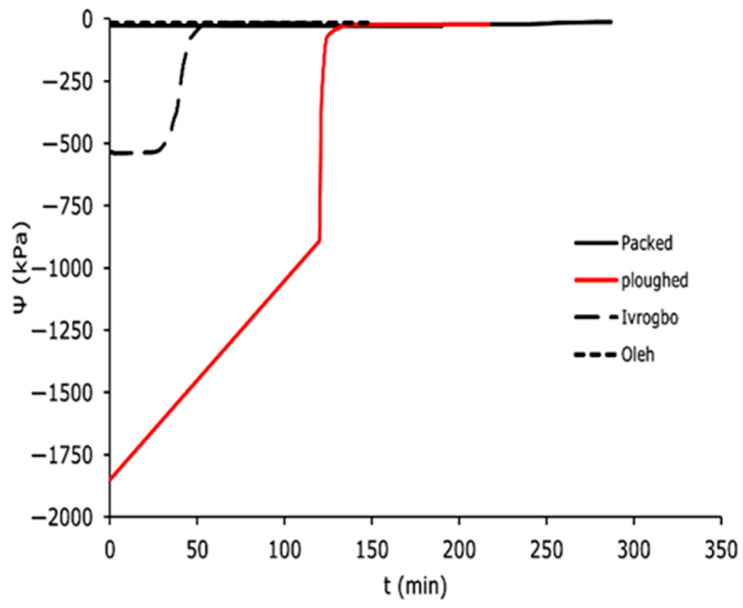
Domain-specific soil matric potential in the soil types at 300 mm.

**Figure 12 sensors-25-04349-f012:**
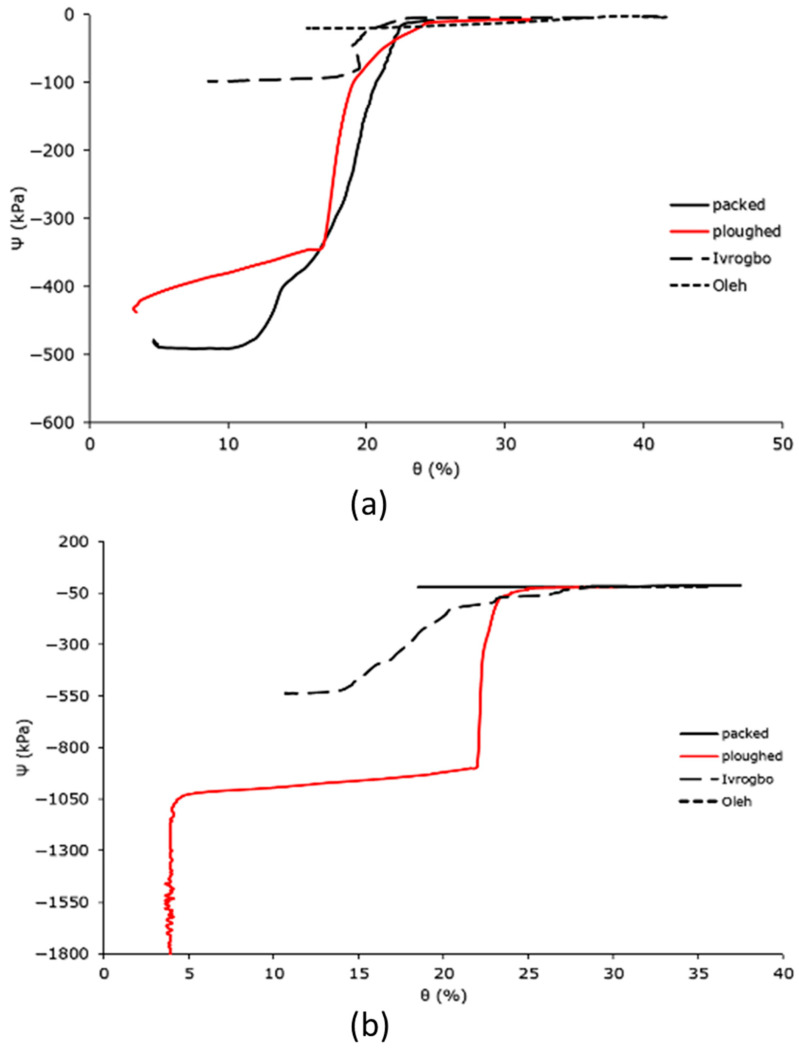
Domain-specific soil water characteristic curves for the different soil types: (**a**) 200 mm; (**b**) 300 mm.

**Figure 13 sensors-25-04349-f013:**
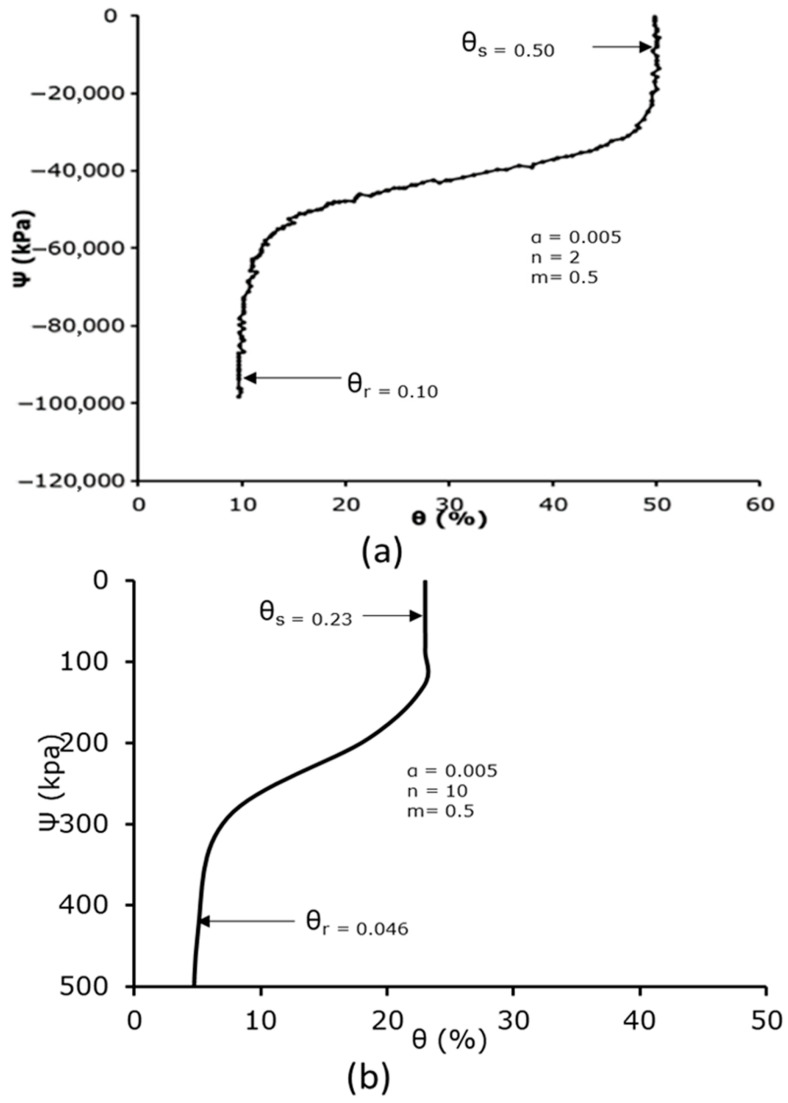
Predicted and measured soil water characteristic curves: (**a**) predicted van Genuchten’s SWCC; (**b**) measured SWCC at 200 mm for the packed soil sample.

**Figure 14 sensors-25-04349-f014:**
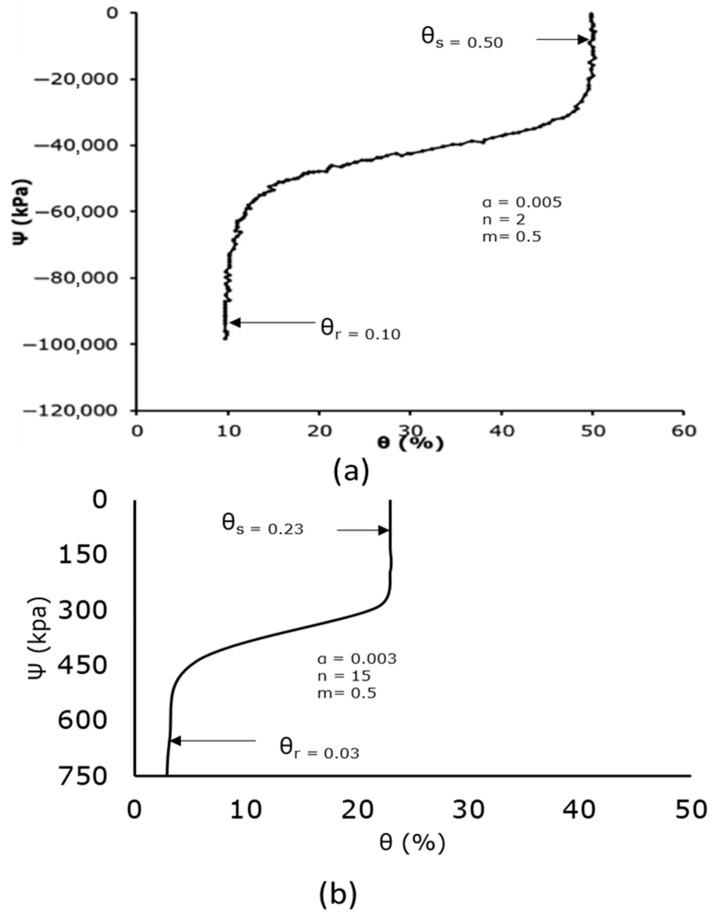
Predicted and measured soil water characteristic curves: (**a**) predicted van Genuchten’s SWCC; (**b**) measured SWCC at 200 mm for the ploughed soil sample.

**Table 1 sensors-25-04349-t001:** Bulk density test.

Type of Material: Lateritic Material	Ivrogbo	Oleh
Container volume (V)	3232	3423
Net container weight	186	188
Container weight + sample	5500	5814
Material weight (W)	5314	5626
Density = W/V	1.64	1.64
Average density	1.64 g/cm^3^	1.64 g/cm^3^

**Table 2 sensors-25-04349-t002:** Gaseous analysis.

Parameters	Ivrogbo	Oleh
Calcium, meq/100 g	4.20	4.86
Magnesium, meq/100 g	2.56	2.47
Potassium, meq/100 g	1.52	1.39
Sodium, meq/100 g	1.98	2.15
TOC, %	0.02	0.03
Total nitrogen, mg/kg	59.27	79.07
Total phosphate, mg/kg	9.69	13.05
Exchangeable acidity, meq/100 g	0.25	0.30
Electrical conductivity, μS/cm	86.30	104.8

## Data Availability

The data presented in this study are openly available at https://doi.org/10.48526/rgu-wt-2795739.

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
