# Peer review of "Hydrodynamic Characterisation of the Inland Valley Soils of the Niger Delta Area for Sustainable Agricultural Water Management"

_sensors, 2025, doi:10.3390/s25144349_

Round 1

Reviewer 1 Report

Comments and Suggestions for Authors

The paper focuses on a laboratory-based experimental design aimed at studying soil water properties. While the manuscript is written in generally correct English, its structure requires significant improvement. The authors present some valuable findings; however, in its current form, the manuscript needs major revisions to be considered suitable for publication as a scientific paper. Below are some suggestions to help improve the manuscript:

  1. The abstract should begin by clearly stating the research problem, followed by the objectives of the study.
  2. The sampling site should be properly described, including its location, climatic conditions, and other relevant characteristics.
  3. The introduction needs to be completely rewritten. The authors should avoid including general lecture-style content on soil physics and instead focus on a critical review of recent and relevant scientific literature.
  4. The study lacks any statistical analysis, which is essential to support the findings.
  5. Based on the experimental setup image, preferential flow may have occurred around the sensors and along the lateral boundaries of the soil column. The authors must justify how they accounted for or mitigated these effects to ensure the validity of their results.

Reviewer 2 Report

Comments and Suggestions for Authors

Ms. Ref. No.: sensors-3714916
Title: Experimental Studies on the Soil Water Characteristics of the Inland Valley Soils of the Niger Delta Area for Efficient Water Management

This study investigates subsurface water flow using soil column experiments and precision sensors to derive soil water characteristic curves for various soil types. Significant variations were observed among samples, with the Van Genuchten model effectively validating the measured soil moisture and matric potential behavior. The present topic is well-suited for the journal of Sensors, although there are a couple of aspects that should be addressed first.

1-The current title seems conventional and not attention-grabbing. I suggest modifying the current title.

2- The main objective of the proposed paper should be clearly stated in the abstract section.

3- Arrange the keywords in alphabetical order and shorten them.

4- Most of the studies referenced in the introduction were published more than ten years ago. It is recommended to include some recent research (published within the last five years) on the topic to enhance the relevance and currency of the literature review.

5- In lines 48–55, the authors emphasize the significance of experimental measurements for assessing soil hydraulic properties. However, the potential role of CFD simulation tools is not addressed, do the authors consider these tools beneficial in this context?

6- The introduction should not include figures or equations; therefore, Figure 1 and the equations (1-5) must be relocated to a more appropriate section within the manuscript.
7-In the introduction section, the innovation and the importance of this work are not clearly highlighted. Please work on this ‎and prove to us why this work is valuable.‎

8- Lines (193-196), the Reynolds number was used to confirm laminar flow conditions, please provide more details on the range of Reynolds numbers observed and how the flow characterization was validated experimentally or analytically?

9-Lines (245-253), regarding the damp packing and drying process used for the Aberdeen sample, justify the choice of six months of air drying and explain how this might affect the soil structure, compaction, and experimental outcomes compared to natural field conditions?

10- add descriptive subfigure titles (captions) for each subfigure (Fig.3.a and b).

11- add descriptive subfigure titles (captions) for each subfigure (Fig.4.a and b).

12- Screenshots should not be used in scientific articles; therefore, Figure 6 must be replaced with a high-resolution image.

13- Line 311, how were the initial soil moisture conditions controlled or standardized across the different soil samples before the infiltration tests?

14- Fig.8, What explains the abrupt dip in water content observed in the packed soil sample after approximately 200 minutes?

15- Figs. 9, 10, and 11, overall, the discussion is lacking in depth and primarily describes the figures without providing the physical significance for each case.

16-Define Nomenclature, Greek symbols, subscripts, superscripts, and acronyms separately in table form.

17-Define abbreviations separately in table form.

18- Conclusions must go deeper; it would be more interesting if the authors focus more on the significance of their findings.

Reviewer 3 Report

Comments and Suggestions for Authors

Reviewer report – “Experimental Studies on the Soil Water Characteristics of the Inland Valley Soils of the Niger Delta Area for Efficient Water Management.”

By Peter Uloho Osame, Taimoor Asim.

The manuscript presents an advanced experimental investigation of the Van Genuchten model of the soil water characteristic curve, aiming to improve the understanding of the complex hydrodynamics of water flow through soil in the context of the inland valleys of the Niger Delta area.

Minor observations:

  1. In the introduction, it should be written about the discrepancies between the estimates of saturated hydraulic conductivity (Ks) obtained using traditional devices (CSIRO-TP, GUELPH-CHP, DRI) and the reference values. Specifically, these discrepancies can be attributed to a combination of factors, such as methodological errors, spatial variability in soil properties, differences in root system density, lack of standardization in experimental procedures, and technological limitations of the instruments.
  2. Redraw Figure 1 for better readability;
  3. Complete the relationship 6 with the following:
  • a is shape parameter related to the air entry value: a=1/hd*(2^(n/n-1))^(1/n)
  • hd is the minimum drainage pressure value. The pressure head at which air begins to enter the soil pores during drainage. It's the pressure at which the soil starts to desaturate from a fully saturated state.

The conclusions are well argued, and the research has significant practical value in the field of agriculture.

A recommendation for future research directions is that the authors could enhance the study by using alternative models, such as the Fredlund & Xing model (which is adaptable to various soil types) or an extension of the van Genuchten, namely, the Durner model (which is ideal for soils with high heterogeneity).

Kind regards,

The reviewer.

Round 2

Reviewer 1 Report

Comments and Suggestions for Authors

The authors' responses were both accurate and perspicacious.